# Public Risk Perception of the Petrochemical Industry, Measured Using a Public Participation Geographic Information System: A Case Study of Camp de Tarragona (Spain)

Edgar Bustamante Picón *, Joan Alberich González, Yolanda Pérez-Albert and Mahdi Gheitasi

Grup de Recerca d'Anàlisi Territorial i Estudis Turístics (GRATET), Department of Geografia, Universitat Rovira i Virgili, 43480 Vila-seca, Spain
* Correspondence: edgar.bustamante@urv.cat

**Abstract:** Following the implementation of the petrochemical industry, the population of Camp de Tarragona began living with a chemical risk, and after two consecutive years of chemical-related accidents with fatalities (in 2019 in the firm Carburos Metálicos, and in 2020 in the firm IQOXE), it is necessary to know the population's perception of this activity. Therefore, this study aims to analyze the population's risk perception regarding the petrochemical industry, by applying a Public Participation Geographic Information System (PPGIS). To this end, the risk perception data obtained from the PPGIS were correlated with the sociodemographic data from the surveys; an analysis was made of the perceived risks associated with this activity and what its possible effect would be on a territorial level, always from the point of view of the surveyed public. The results showed that the population clearly identifies on the map what the main sources of chemical risk are and which places would be affected by a possible explosion. In addition, it was verified that certain sociodemographic characteristics, such as gender or age, affect people's perception of the risk, and that the weather conditions and anomalous situations within the industry also influence people's perception, indicating high values of danger.

**Keywords:** risk perception; technological risk; PPGIS; petrochemical industry; Tarragona

## 1. Introduction

The petrochemical industry specializes in transforming hydrocarbons into chemical products, and it is grouped into large cluster complexes that need significant investments and overland extensions for the manufacturing facilities. These industries, largely in the hands of multinational companies, manufacture their production coordinately, concentrated in the territory and often associated to port facilities.

Industrial petrochemical production carries all kinds of risks. Those appearing most frequently in scientific literature refer to the long-term risks triggered by the effects caused by the population living alongside these industrial estates for an extended period of time, which can affect the health of both humans and animals (respiratory problems, cancer or other issues associated with health) and the health of the environment (air, water and land pollution). Out of the most studied effects on the human population, there is the question of whether prolonged exposure to a series of pollutants given off by the petrochemical industry is associated with a higher incidence of cancer and a greater mortality rate for this same cause (see the bibliographical data compilation by Domingo et al. [1]). Additionally studied are the prevalence of other illnesses, such as asthma and other respiratory problems, both in the child and adult populations who live near petrochemical complexes, as well as the reproductive results in pregnant women (consult the bibliographical review by Marqueès et al. [2]). On the other hand, the short-term risks, which refer to the immediate, catastrophic effects after an incident or accident, and which also influence the population's mental and physical health, have been studied to a lesser degree. This article will focus

on these types of risks and the population's perception of them. In order to define the risk perception, the ideal approach is to deal with the different aspects separately.

Risk is the probability of suffering damage [3], whether this damage is caused to material goods, the environment or living beings. Casal and Vílchez [4] consider that an individual risk appears when people are exposed to a hazard 24 h a day, 365 days a year. In this case, the study comprises all the residents near one of the petrochemical complexes in Camp de Tarragona. This means we have to analyze an anthropic–technological type risk, which results from handling hazardous products with advanced technologies, and which can lead to damage caused by human intervention, voluntary or not, as a result of not respecting the safety regulations or the principles aimed at governing the production, transport, handling and storage of certain products. This compromises the necessary balance that should exist between the communities and the environment [5].

The General Directorate of Civil Protection and Emergencies of the Generalitat de Catalunya [6] defines chemical risk as that which is caused by the establishments that store, manufacture and/or handle large amounts of hazardous substances that can cause serious accidents, and which exceed the limits of the chemical facilities themselves and affect the immediate surroundings, the environment and goods. It has been proven that chemical accidents occurring near densely populated areas can cause colossal damage, and so chemical accidents near a populated area can be potentially catastrophic with a high number of victims. The general public's proximity to these chemical facilities has been identified as a significant cause of increased human exposure in 43% of the accidents investigated by the U.S. Chemical Safety Board (CSB) [7].

According to García [8], perception is an individual cognitive process that is supported by each person's information on different issues (experiences, memories, feelings, external information, etc.) and which the brain processes immediately and organizes into an opinion or value.

However, in order to get a more detailed idea of what risk perception is for different people, it is necessary to take into account other no less significant terms that contribute to this personal perception. These additional terms are as follows:

- Delusional optimism. This occurs when there are no previous negative experiences and there is a lack of information. In this case, the person affected by delusional optimism thinks that the negative episodes that may occur, will happen to other people or in other places, but never to them [9].
- Acceptable risk. Individuals weigh up possible damages and benefits, and establish an acceptable risk limit, whereby the greater the benefit implied by being next to a possible risk, the greater they will accept this risk. For example, an individual who works in a petrochemical complex will have greater risk acceptance than another individual who lives a few meters away and receives no benefit from it [10,11].

Although the aim of this research is to know the risk perception of the population living near petrochemical complexes, as indicated by Grant [12], it must be taken into account that people are not a good source for assessing risks, and therefore, fail in their perception of them for the following reasons:

- False feedback. The fact that industrial accidents occur so infrequently means that most people have never experienced one, and so an expectation arises whereby every day passes without any accident, and so people are convinced that the activity will not cause them any harm.
- Cognitive interference. The shortcuts that the brain uses influence the way that risks are recognized. The brain is prepared to process large amounts of information in a fast and very sophisticated way. One of the mechanisms that helps this is the brain's capacity to compensate for deficiencies and recognize patterns. For example, when reading, people do not concentrate on the individual letters in each word; instead, they capture the meaning of the word as a whole. This happens because the context and the initial clues help us to leap forwards to what we want to see. This capacity, which is generally very useful, can be an impediment when it comes to recognizing

risks. When a person finds themselves in common situations, they tend to see what they expect to see, and if the situation has changed in any way which introduces a risk, it is possible that the person does not realize this modification. This usually happens when driving along a known route, where a new stop sign has been set up. Drivers often do not see the sign, even though it is in full view.

- Being used to things. When surrounded by an activity such as industrial production, there are two factors that influence our safety: experience (which provides the knowledge on how to be safe in the various possible circumstances) and awareness (which makes us quickly recognize the circumstances in which it is necessary to react). When a person is new around the activity, they are very aware, but have little experience. These people are very sensitive to risk; however, they do not have much capacity to deal with it. As time passes, experience grows, but awareness diminishes as the activity becomes less new, or less of a novelty, and more of a routine experience. Over time, people have more experience and less awareness, and in this case, it is less likely that risk is recognized. In short, people become insensitive to dangers.

As indicated in the literature, it has been proven that when an individual speculates about the events that may occur to them in the future, they allow themselves to be invaded by an unfounded optimism regarding the negative events and positive events that may occur. However, the risks only suggest what should not be done, not what should be done. Once risks become the backdrop for everything, the alarm triggered creates an atmosphere of impotence and paralysis. Whether doing nothing or doing too much, the world becomes a series of indomitable risks. We could call this the risk trap, which is what the world can look like through the prism of risk perception [13].

Different methods exist for analyzing the risk perceived by the population. One that is widely applied is the Public Participation Geographic Information System (PPGIS); these are tools that were created in 1996 to support the involvement of local communities in spatial planning [14]. They help to gather georeferenced information by non-expert users [15], combining surveys with mapped information and are able to map and analyze the vulnerability of a wide range of landscape or environmental values, inter alia [16], taking into account the population's sociodemographic characteristics [15]. In their early days, the PPGISs were based on simple tools, such as paper maps, pens and stickers. Later, digital mapping using computers was introduced, first as an isolated approach (offline) and, later, using the Internet (online) [14].

The central principle of this method Is that it allows participants to directly identify the locations they want to value and to digitalize them in a GIS database for spatial analysis [16]. The PPGIS' features make them a valuable tool for territory administrators and planners, since for them it is essential to take into account the population's opinion and perception [15]. This idea guides the development of one of the most important aims of the PPGIS, which is involving citizens in the planning processes. This implies involving them in the decision making on issues directly affecting their lives [14], which reflects what they perceive in their physical environment according to their ideas [17].

Risk analysis, as in this case, has an obvious geographic dimension and the PPGISs are an appropriate solution for addressing and going further into this type of study that requires the population's involvement and assistance. Nevertheless, only a few studies focus their analysis on technological type risk perception based on the public participatory process (particularly in the petrochemical industry), with a spatial type vision, which, as well as revealing the degree of risk that the population perceives, also works on its spatial dimension. For example, Di Fonzo et al. conducted a review of existing research on the distribution of health hazards from industrial pollution and discovered a significant relationship between the social dimension and industrial contamination health hazards [18]. Dettori et al. [19] presented a study that aimed to assess risk perception and community outrage related to environmental factors among a self-selected sample of citizens living in an area with high emotional impact industrial structures. All of the preceding studies used interviews, surveys and other methods to examine risk perception, but none of them used

PPGIS as a methodology or took into account the spatial dimension. As a result, failing to consider a PPGIS for the analysis of perceived risk in the petrochemical industry may be regarded as a knowledge gap.

Once this knowledge gap is established, the main aim of this work is to determine the population's perception of the petrochemical risk and relate it to sociodemographic characteristics and different meteorological and industrial production situations. In the specific case of this study, the territorial reference scope is Camp de Tarragona (Catalonia, Spain), an area that is home to one of the main petrochemical industrial complexes in southern Europe. This aim is broken down into three sub-aims. The first focuses on designing an online survey that includes a series of space-based questions that allow the residents or workers in the indicated territorial scope to locate on a map the main risk emission points, and the areas that can be affected by them. The second intends to determine a sociodemographic characterization of the informants in order to capture the different risk perceptions according to the different social and demographic groups the people belong to, with particular emphasis on the differences between gender and age. In this respect, constructing a logistic regression model will reveal the variable with the greatest impact on the existence of high-risk perception. The third sub-aim is to know what influence environmental conditions have, for example, between day and night, between a cloudy day and a sunny day, and between the feeling of safety or risk arising from the petrochemical activity.

The article is divided into five sections, including this one, plus the bibliography. Section 1 reflects on the basic concepts for understanding the relationship between the petrochemical industry, the population's health and establishing the risk perception, as well as addressing the use of a PPGIS to analyze it and establish the aims of the study; Section 2 describes the scope of study; Section 3 explains the methodology and the tools used; Section 4 focuses on the results obtained; and Section 5 contains the discussion and conclusions.

## 2. The Territorial Context

The case study focuses on Camp de Tarragona, an area located in the northern end of the province of Tarragona, along the Spanish Mediterranean coast (Figure 1). The region has a population of more than 500,000 inhabitants. Tarragona and Reus, both with over 100,000 inhabitants, are the area's main cities and a continuum of costal urban settlements connects them. This is an interesting area for several reasons. First, it includes several urban forms, from sparsely populated, low-density areas to dense and diverse urban centers. Secondly, there is a unique combination of intensive tourism and industrial activities: the coastal towns in the area make up what is known as the Costa Daurada, which is one of the most visited tourist destinations in Spain, and these towns share the area with one of the largest petrochemical complexes in western Europe, the Universitat Rovira i Virgili (with a campus in the area and more than 13,000 students) and other, less important economic activities, such as agriculture.

The coastal area of Camp de Tarragona represents a clear example of space that has suffered serious disorganization in a short period of time, concerning some fast processes of industrialization and town planning not subject to an appropriate master plan. The petrochemical refinery and other auxiliary companies arrived in Camp de Tarragona in two stages: the first between 1965 and 1970 (Polígono Entrevías de Tarragona—Polígono Sur) and between 1971 and 1980 (La Pobla de Mafumet and its vicinity—Polígono Norte). Implementing a pollutant activity has caused some significant environmental problems and, consequently, a reduction in the population's quality of life (pollution, noise, smells, etc.) [20]. In fact, in response, the population in La Pobla de Mafumet filed a formal complaint because of the foul odors emanating from one of the new plants located in the refinery [21].

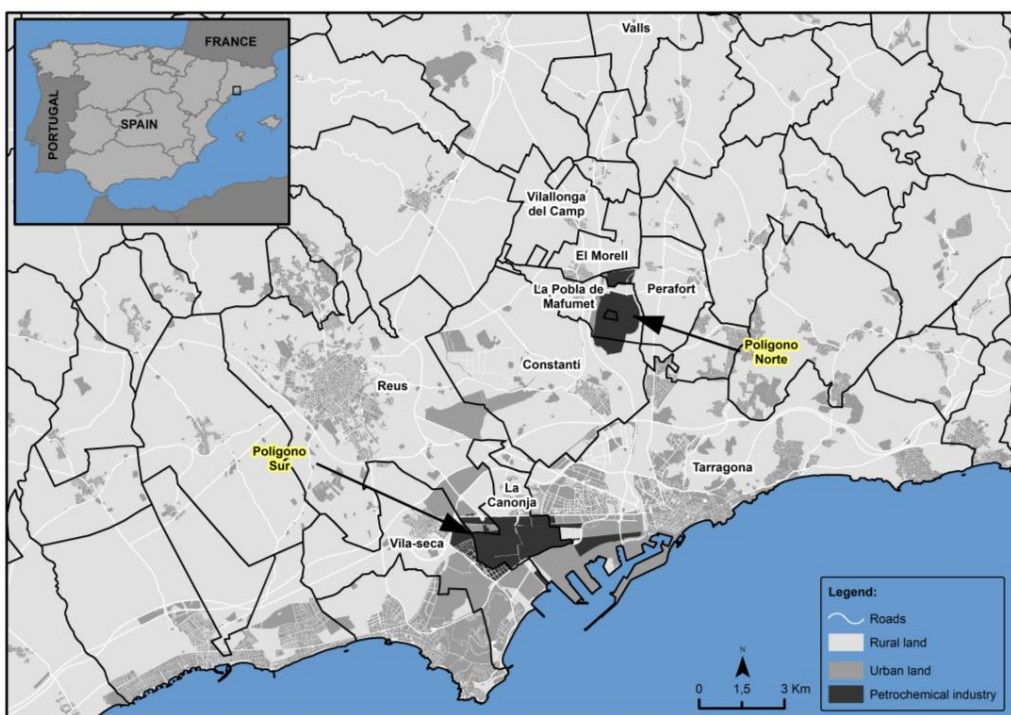

**Figure 1.** Localization of the study area.

The three fundamental facts that have driven the arrival of this industry are as follows: first of all, the proximity to the port of Tarragona, which facilitates importing the necessary petrol; secondly, the existence of many transportation routes, both on the roads and railways (train lines to Tortosa, Reus and Roda de Berà); thirdly, the possibility of using the plentiful water from the Ebro river during the production processes. The gradual expansion and diversification of the facilities led to intense industrial dynamics, which meant intercompany productive integration and a wider market perspective, which in turn made it easier to set up new companies, to the point that this area is one of the main petrochemical industrial estates in Southern Europe.

At present, the petrochemical activity is developed in two clearly different industrial estates: the so-called Polígono Norte (northern estate) covers a surface area of 470 hectares in the towns of La Pobla de Mafumet (Figure 2), El Morell, Perafort and Constantí; while the Polígono Sur (southern estate), in the towns of Tarragona, La Canonja and Vila-seca, occupies about 720 hectares. Both estates are home to about thirty companies; in the northern area, the companies specialize in processing the raw material (refining the petrol to obtain liquid pressurized gas, diesel oil, kerosene, naphtha, propylene and ethylene, among others) and in the southern area, they specialize in preparing the final product based on the processing done in the northern area (acids, alkaline salts, fertilizers, insecticides, fuels, plastics and synthetic essences, among many other products).

Despite its undeniable role in the economic activity and employment in the region, the industrial implementation in the area is the object of a certain amount of rejection by society, not only because of the emission of pollutant gases, large flames and bad smells, but also because of the constant incidents and accidents of varying severity. The most recent, in January 2020, caused three deaths: two workers in the firm where the explosion occurred and a local neighbor, whose house was hit by a large part swept up by the explosive wave.

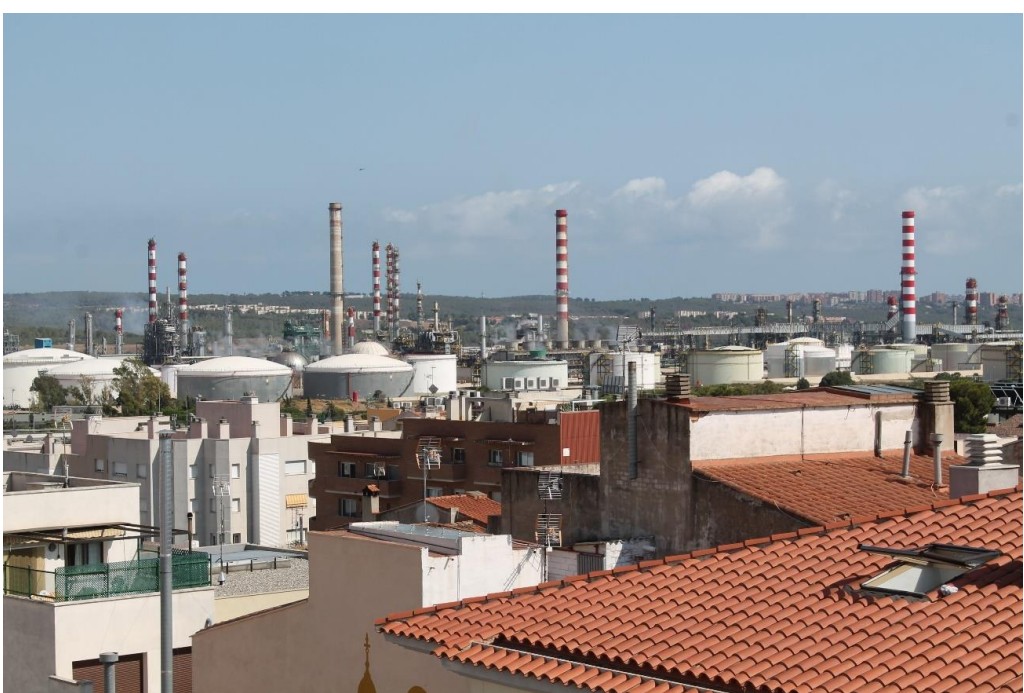

**Figure 2.** Polígono Norte from the nearest village (La Pobla de Mafumet).

## 3. Methodology

### 3.1. The Survey, the PPGIS

The PPGIS designed for this research was hosted on the surveymonkey.com digital platform. It was possible to access it via computer, but not from other electronic devices such as smartphones or tablets due to the requirements of the space-type questions based on Google maps inserted into the actual question, and where you have to reply by dragging the available icons to the desired location. This information was available before starting the survey, and participants were also informed about the treatment of the data (as can be seen in Figure A1). Participants were not provided with information about the subject matter of the study in order not to influence their responses. To facilitate participation, two versions of the survey were prepared, one in Spanish and another in Catalan. It contained 26 questions structured into two sections: sociodemographic data and risk perception. The survey was circulated via email newsletters, links in local press, websites, and social media, university, local councils, environmental NGOs, all factories in the petrochemical industrial estate, and so on.

In the first block of sociodemographic questions, apart from the usual ones on gender, age, level of education or employment status, others were added, such as the place of residence, work or study (indicating this on an interactive map, as shown in Figure 3 or indicating the postcode), in order to assess whether the distance influences risk perception. Additionally, the respondent was asked whether they have worked at any time in the petrochemical industry or if any direct family member has and, finally, whether from their place of residence they have direct views towards some of the petrochemical complexes in the area.

In the second block (risk perception), each respondent was asked about their personal risk perception on a general level (Likert scale of 1 to 5 where 1 indicates lower risk perception and 5 greater perception), their risk perception in different environmental situations (day, night, sunny day or rainy day) and in different anomalous situations within the factories (flames and smoke in the torches, loud noises and smells of chemical products) and about the type of risk (explosion, fire, toxic cloud, all the above, others or no risk) they believe is more likely to occur. Likert scales were used in this study as the familiar five-point bipolar response that most people are familiar with. These scales, which range

from least to most, ask people to indicate how much they agree or disagree, approve or disapprove, or believe to be true or false [22].

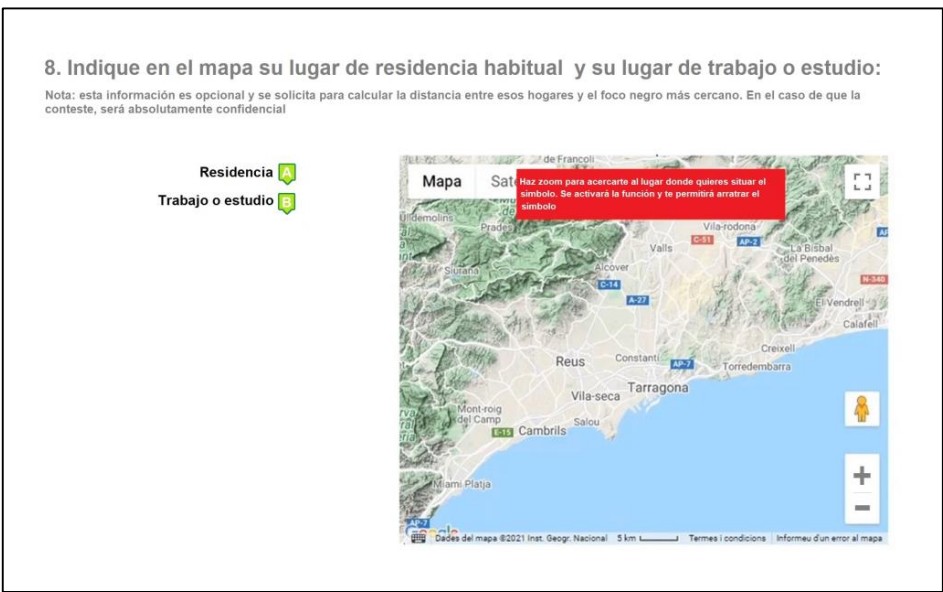

**Figure 3.** Example of a map-based question included in the survey. Translation: Indicate your usual place of residence and your place of work or study: Note: this information is optional and is requested in order to calculate the distance between these places and the nearest black spotlight, in case you answer it, it will be absolutely confidential. Icon A: Residence. Icon B: Work or study.

Two spatial questions were enclosed in this block. In the first one, respondents had to locate on the map the sources of petrochemical risk and assess their hazardousness. In the second one (Figure 4), the places that would be affected by possible risks and their intensity.

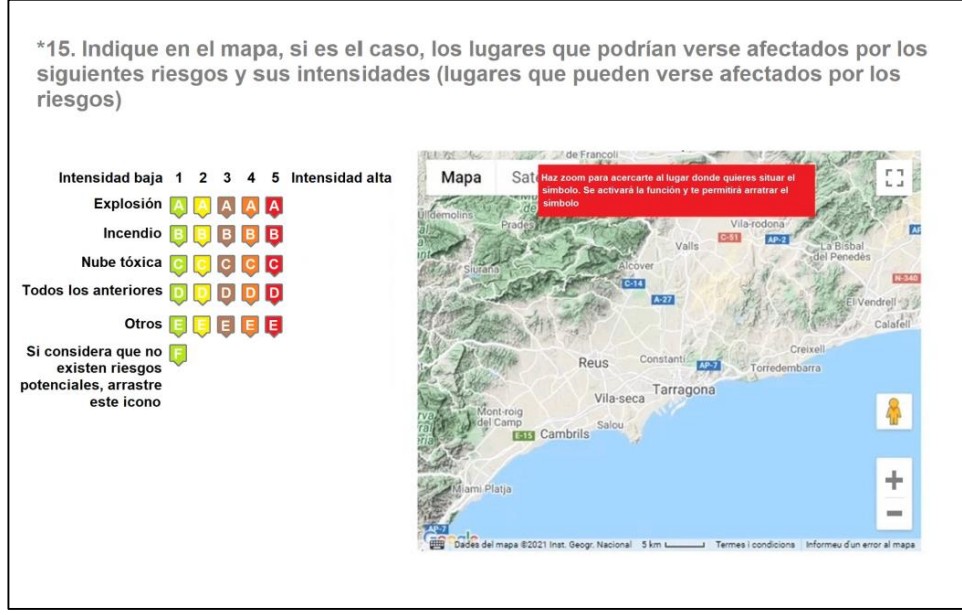

**Figure 4.** Map-based question on the impact of possible risks in the area. Translation: Indicate on the map, if applicable, the locations that could be affected by the following hazards and their intensities (locations that could be affected by the hazards). Legend: Low intensity 1 2 3 4 5 High intensity. Icons: Explosion, Fire, Toxic cloud, All of the above, Other, If you consider that there are no potential hazards, drag this icon.

### 3.2. The Statistical and Spatial Analysis

The results from block 1 (sociodemographic characteristics) were processed in SPSS by applying basic techniques of descriptive statistics (frequencies) and causal analysis (bivariate correlations and logic regressions).

In order to measure the possible degree of mutual influence among these variables, a model was built based on developing logistic regression, showing the net impact of each of the independent variables (in this case, the sociodemographic characteristics of the informing public) over the dependent (the degree of petrochemical risk perception).

Owing to the characteristics of logistic regression, the different categories of one and the same variable are processed separately, so that the differences observed between each category and the rest are analyzed. This analysis can be carried out in two ways: either all the categories are compared with a reference category or the reference category is the set, which is the option chosen here. Thus, the interpretation to be made of the odds ratio (expβ) in a specific category is the incidence of having a high level of risk perception, compared with the incidence in the surveyed population as a whole. Specifically, each odds ratio indicates how different this relationship is, if the variable takes that value, with respect to the population as a whole. In these terms, if the value is above unity, it indicates how many times higher the probability is that someone in that category has a higher risk perception than the average. Similarly, and symmetrically, it is necessary to read the values under unity. Equally, given the magnitude of the sample, emphasis should be placed on those where the odds ratio ($\alpha < 0.001$) has statistical importance.

Several logistic regressions were carried out: first of all, a set of regressions which have been called individual models, which correspond to bivariate regressions, since the incidence of each independent variable over the dependent is assessed separately. Secondly, the joint model or multivariable model, to which all the variables have been added, whereby it is necessary to read their results as the contribution of each one irrespective of the rest. Specifically, in this last case, the so-called stepwise regression has been used, which, generally speaking, consists of entering the variables one by one, and assessing with each new entry whether any of the variables entered ceases to have a significant contribution and, in this case, exclude it. At the same time, the order in which the independent variables are included in the model gives an idea of their contribution to the explanation of the variability of the dependent variable.

Overall, a total of eight independent variables were entered into the model, in an attempt to guarantee representing the traditional demographic variables (gender and age), some socioeconomic and employment ones (level of education and employment or not in the petrochemical industry) and, finally, the territorial variables (years residing near the industrial estate, direct visibility towards the industry and the differences between the place of residence and the place of work/study vis-à-vis the petrochemical activity in the industrial estate). Two of these variables (age and distance) are a continuous type, while the other three (gender, level of education and employment in the petrochemical sector) are categorical type variables. In the specific case of the level of education, this has been re-categorized, on the one hand, to guarantee a certain number of cases in each category and, on the other hand, to group certain categories which previously had shown similar behavior to the dependent variable. Furthermore, in the case of the categorical variables, it is necessary to previously define the reference category, so that the odds ratio indicates the degree and sign of the risk perception of the rest of categories of that variable with respect to what has been taken as reference. The percentage distribution of the sample according to each variable included in the logistic regression models is shown in Table 1.

**Table 1.** Percentage distribution of the sample according to each variable entered in the logistic regression models.

| Type of Variable | Variable | Category | % |
|---|---|---|---|
| Demographic | Gender | Man * | 47.6% |
| | | Woman | 52.4% |
| | Age | 16–24 years | 35.5% |
| | | 25–34 years | 23.2% |
| | | 35–44 years | 16.2% |
| | | 45–54 years | 13.7% |
| | | 55–64 years | 8.6% |
| | | 65 years and older | 2.8% |
| Socioeconomic/ employment | Level of education | Secondary studies * | 13.5% |
| | | Professional training | 9.3% |
| | | University studies | 77.2% |
| | Employment in the petrochemical industry | Yes * | 68.9% |
| | | No | 31.1% |
| Territorial | Visibility of the petrochemical industrial estate | No * | 61.5% |
| | | Yes | 38.5% |
| | Years living near the petrochemical estate | Does not live near * | 8.6% |
| | | Less than 10 years | 17.4% |
| | | From 10 to 19 years | 19.3% |
| | | From 20 to 29 years | 26.5% |
| | | 30 years or more | 28.3% |
| | Distance to place of residence | Less than 5 km | 26.9% |
| | | Between 5 and 10 km | 56.6% |
| | | More than 10 km | 16.5% |
| | Distance to place of work/study | Less than 5 km | 17.6% |
| | | Between 5 and 10 km | 64.3% |
| | | More than 10 km | 18.1% |

Note: in the categorical variables, the asterisk (*) indicates which is the category taken as a reference.

The results from the spatial questions were imported into a Geographical Information System (GIS), specifically ArcMap (v. 10.3). To take into account the assessments as a whole, the accumulated risk perception was calculated in each place (pixel). Therefore (1) a layer was created by rastering all the opinions, excluding the values 0 or No Data; (2) all the assessments in one and the same range (1 to 5) were added together, obtaining the frequency of each value; (3) each value was multiplied by its frequency and (4) the values were standardized from the raster calculator by applying the following formula proposed by Passuello et al. (2012):

$$z = (x - \mu)/\sigma \qquad (1)$$

where:

- $\mu$ is the average of the values;
- $\sigma$ is the standard deviation of the values;
- x are the values that are indicated in the observation.

The 'Hot Spot' function has been applied over this layer, and the hot points and cold points were obtained according to the opinions gathered in the survey.

## 4. Results

### 4.1. The Territorial Distribution of the Risk Emission Sources and the Affected Areas

Once all the coordinates that the population had entered in the PPGIS were georeferenced, a total of 5317 opinions were obtained from 431 participants, which are distributed as shown in Table 2.

**Table 2.** Distribution of the georeferenced points.

| Type of Information | Type of Risk | Points | Total |
|---|---|---|---|
| Sources of risk | Intensity 1, 2, 3, 4 or 5 | 1801 | |
| | There are not sources of risk | 46 | 1847 |
| Places affected | By explosion | 808 | |
| | By fire | 387 | |
| | By toxic clouds | 1060 | |
| | By all the above | 1065 | |
| | By other causes | 106 | |
| | There are no places affected | 44 | 3470 |
| Total | | | 5317 |

The territorial distribution of the petrochemical risk emission sources that the users identified is shown in Figure 5. The color of each point indicates the degree of perceived danger, as well as the spatial relationship established between them: the warm colors (from yellow to red) indicate those points with high danger scores that are located near similar points, while the colder tones (blue range) show the points with a low level of danger surrounded by points with similar characteristics.

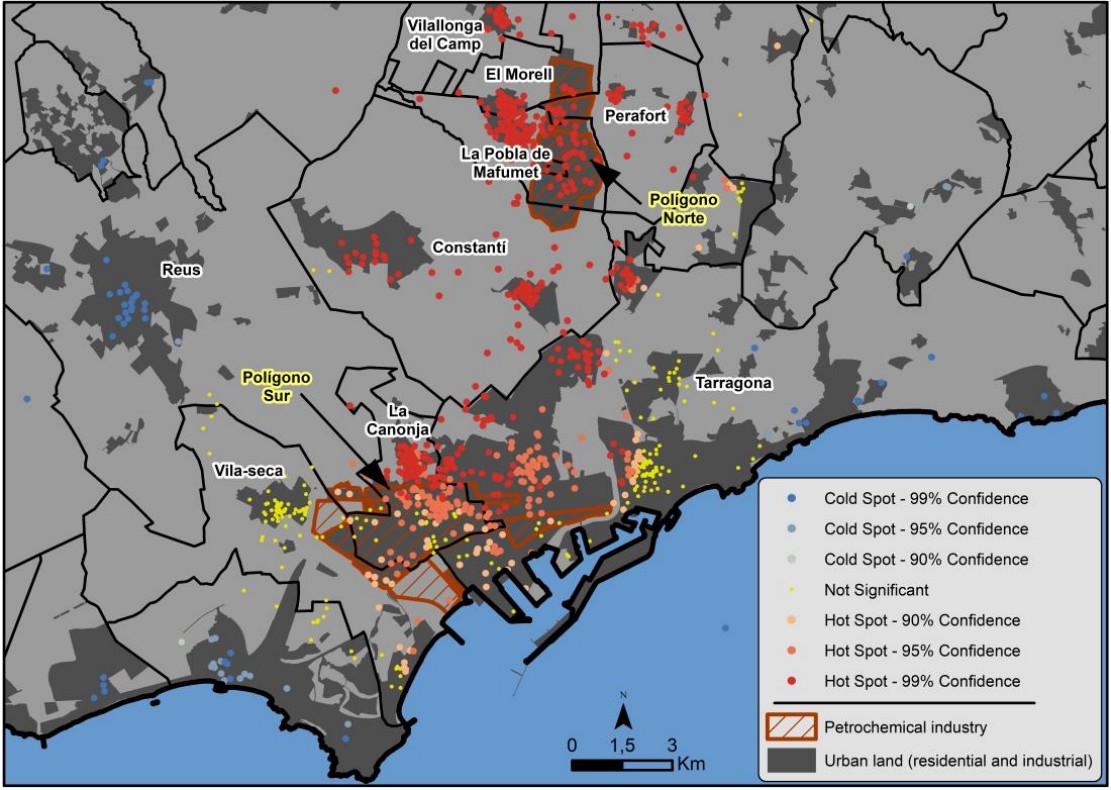

**Figure 5.** Territorial distribution of the petrochemical risk sources and perception of their degree of hazardousness.

Most of the risk sources are within the perimeter of the two petrochemical estates, with the highest concentration of emission sources being in Polígono Norte. On the other hand, it is somewhat surprising that this concentration is also seen in the town districts nearest the estates (Tarragona, El Morell, La Pobla de Mafumet or Perafort), and even in some further away (Vilallonga del Camp, for example) where, in principle, there are no activities that imply the risk of an industrial accident. This is not the case of the situation perceived in the center of the town of Reus, where, as expected, there is a concentration of points of low perceived danger.

The map of the territorial distribution of the affected areas in the event of a petrochemical accident (Figure 6) shows certain similarity to the map of the emission areas: the points with the highest risk are distributed both inside the industrial estates themselves and in the nearest town centers. Particularly relevant is the distribution of points that would be affected by a hypothetical explosion, which is, without a doubt, the perceived risk with greatest territorial repercussion (Figure 6a). An opposite situation is that of the possible areas affected by fire, perceived as much more territorially limited risk (Figure 6b). Finally, it is worth highlighting the high number of opinions that identify locations that would be affected not only by a particular type of risk, but also by the total risk (Figure 6f). Again, it is worth pointing out that Reus and its nearest city environment, due to its peripheral position with respect to the areas with greater industrial concentration, is perceived as a safe space vis-à-vis most of the risks (Figure 6).

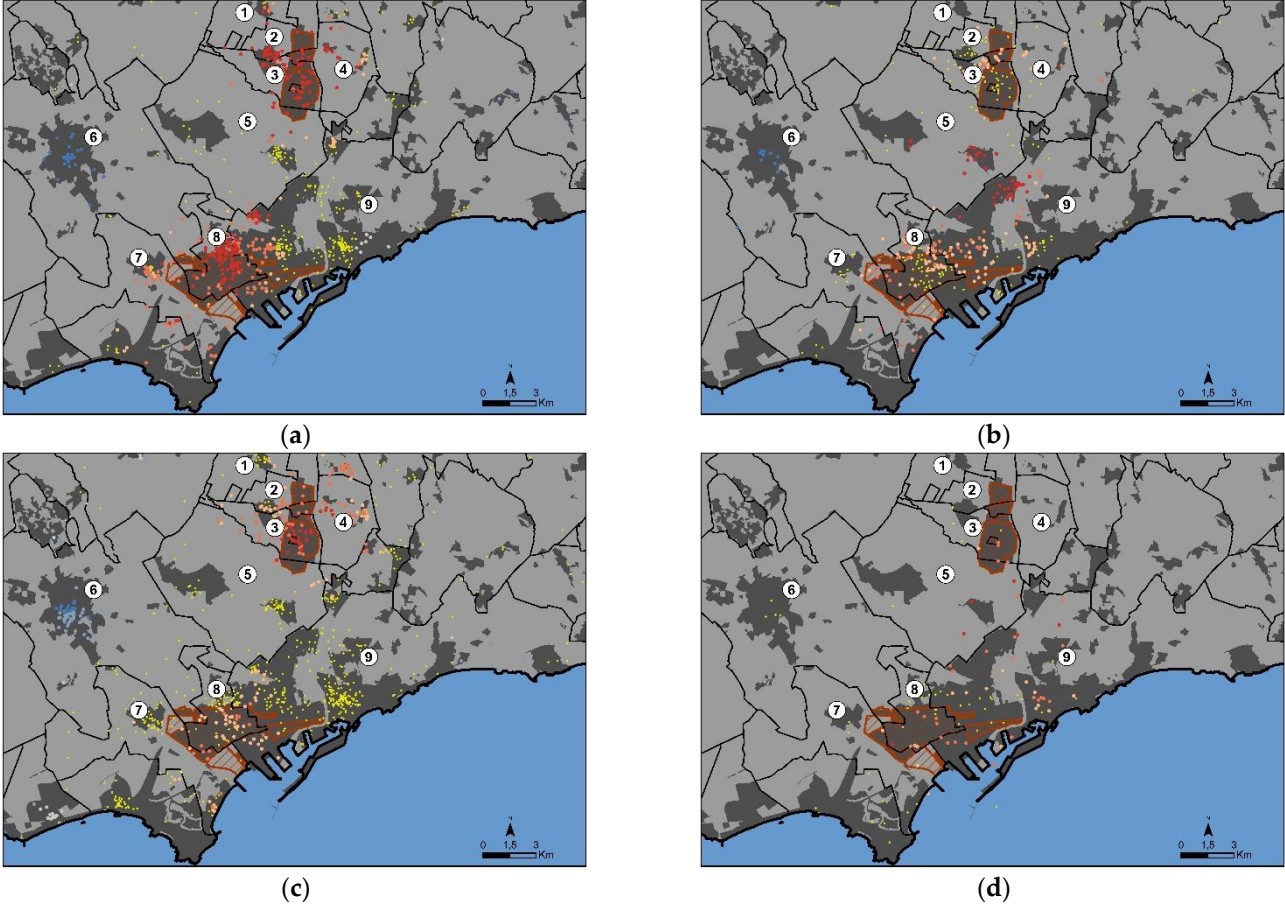

(**a**)

(**b**)

(**c**)

(**d**)

**Figure 6.** *Cont.*

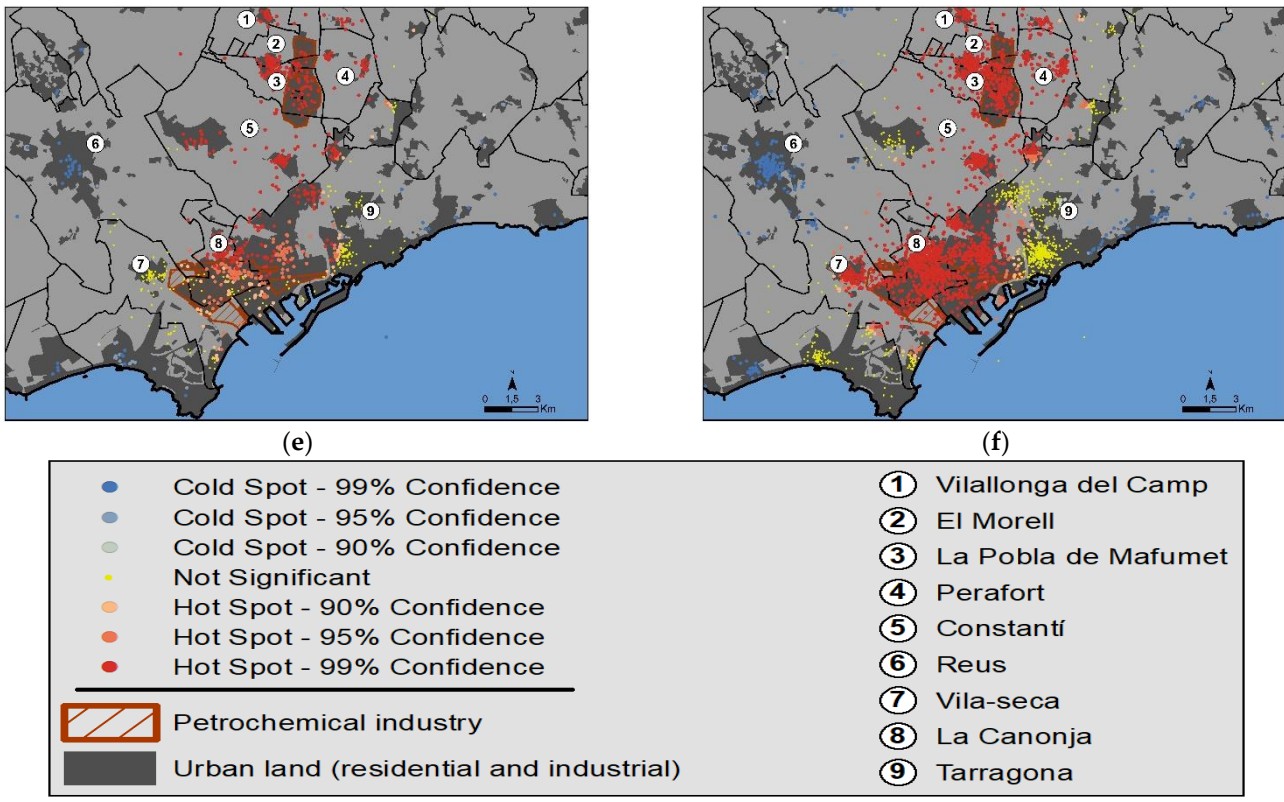

(e)  (f)

| | |
|---|---|
| ● Cold Spot - 99% Confidence | ① Vilallonga del Camp |
| ● Cold Spot - 95% Confidence | ② El Morell |
| ● Cold Spot - 90% Confidence | ③ La Pobla de Mafumet |
| · Not Significant | ④ Perafort |
| ● Hot Spot - 90% Confidence | ⑤ Constantí |
| ● Hot Spot - 95% Confidence | ⑥ Reus |
| ● Hot Spot - 99% Confidence | ⑦ Vila-seca |
| ▨ Petrochemical industry | ⑧ La Canonja |
| ▮ Urban land (residential and industrial) | ⑨ Tarragona |

**Figure 6.** Territorial distribution of the areas that would be affected in the event of a petrochemical accident according to the type of risk and its degree of hazardousness. (**a**) Explosion; (**b**) fire; (**c**) toxic cloud; (**d**) other risks; (**e**) all the risks; (**f**) set of risks.

*4.2. The Level of Perceived Risk and the Sociodemographic Characterization of the Population*

While the exhibition of results to date was based on the territorial perspective, locating the possible risk sources and the places that would be affected by them, it is equally interesting to relate the overall risk perception with the sociodemographic characteristics and the distance to the petrochemical facilities.

A first approximation to the data (Figure 7) shows that nearly two-thirds (63.0%) of the people surveyed stated they had a high level or a very high level of risk perception, as opposed to the 13.2% who stated that their perception level was low or very low.

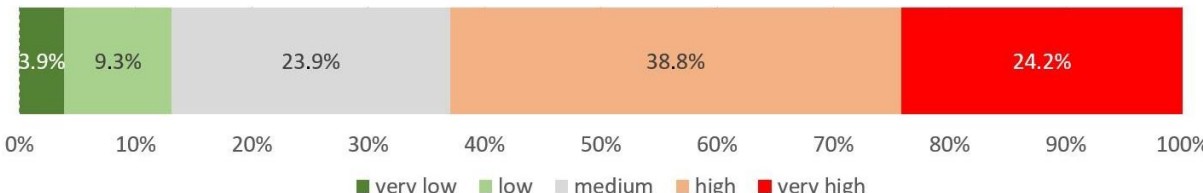

**Figure 7.** Perception level of the petrochemical risk.

However, the degree of risk perception of the surveyed population as a whole differs according to their sociodemographic characterization and their location of residence and/or work. The analysis of each of these variables (gender, age, level of education, employment status, the fact that they work in the petrochemical sector, distance from the place of residence and/or work or study and the visibility of the petrochemical facilities from the place of residence) was conducted from two complementary viewpoints. Firstly, by means of bivariate correlations of each one with the stated declared risk, where the former act as independent variables and the latter, as a dependent variable. Secondly, in order

to complete this simple correlation between variables, it was decided to build a more complex model, by creating a logistic correlation, which measures the joint incidence of these independent variables on a possible high-risk perception.

Several results can be drawn from the demographic variables (Table 3). First of all, regarding gender, women have a greater perception of the petrochemical risk than men: while their percentage of high perceptions is 58.9%, among women this value reaches 66.8%. With respect to age, there is a clear gradation, as the level of worry increases with age. Closely related to this fact is the perception linked to the level of education: due to the influence of the age-based structure in each training level analyzed, the least concern appears among people with a lower level of education (which, in turn, is linked to the older population, but, particularly, in this case, the youngest population that has had greater opportunities to access education), while, among the top levels of education, the level of concern is much higher. This data confirms that the students have the lowest risk perception indices, compared to the perception of workers and, above all, retired people. The fact that the person surveyed or one of their direct relatives works in the petrochemical sector influences risk perception: in the affirmative cases, only half the informants perceive a high risk (which leads to an average perception value of 3.39, the lowest out of all the groups analyzed), while, in the opposite cases, this percentage rises to 67.5%.

**Table 3.** Distribution of the surveyed population according to their perception level of the petrochemical risk and its demographic variables.

| Variable | Category | Perceived Level of Risk | | | | | |
| --- | --- | --- | --- | --- | --- | --- | --- |
| | | Very Low (1) | Low (2) | Medium (3) | High (4) | Very High (5) | Average |
| Gender | Man | 4.7% | 11.6% | 24.7% | 32.6% | 26.3% | 3.61 |
| | Woman | 3.0% | 7.0% | 23.1% | 44.7% | 22.1% | 3.75 |
| Age | <30 years | 3.1% | 9.9% | 39.7% | 37.5% | 19.8% | 3.59 |
| | 30–39 years | 8.5% | 8.5% | 16.9% | 42.4% | 23.7% | 3.68 |
| | 40–49 years | 2.9% | 8.7% | 26.1% | 34.8% | 27.5% | 3.74 |
| | 50–59 years | 5.3% | 5.3% | 7.9% | 47.4% | 34.2% | 3.90 |
| | 60 years and more | 0.0% | 12.9% | 16.1% | 38.7% | 32.3% | 3.91 |
| Level of education | Primary or secondary | 3.3% | 15.0% | 36.7% | 33.3% | 11.7% | 3.35 |
| | Professional training | 7.5% | 5.0% | 20.0% | 30.0% | 37.5% | 3.85 |
| | University education | 3.6% | 10.0% | 22.4% | 38.4% | 25.7% | 3.73 |
| Employment status | Student | 5.2% | 9.8% | 31.0% | 36.8% | 17.2% | 3.51 |
| | Unemployed | 11.1% | 0.0% | 33.3% | 11.1% | 44.4% | 3.78 |
| | Employee | 3.4% | 10.7% | 18.5% | 38.0% | 29.3% | 3.79 |
| | Freelance | 0.0% | 13.6% | 22.7% | 31.8% | 31.8% | 3.82 |
| | Retired—Pensioner | 0.0% | 10.0% | 15.0% | 45.0% | 30.0% | 3.95 |
| Work in the sector | Yes | 6.8% | 15.8% | 27.1% | 32.3% | 18.0% | 3.39 |
| | No | 2.7% | 7.8% | 22.0% | 39.3% | 28.1% | 3.82 |
| Total | | 3.9% | 9.3% | 23.9% | 38.8% | 24.2% | 3.68 |

Another group of variables which, in principle, would have to be taken into account when analyzing the degree of perception of the petrochemical risk are the territorial and spatial type variables (Table 4). However, the results are inconclusive in this respect: first, there is no evidence of a differential assessment of the petrochemical risk depending on whether, from the place of residence, there is the possibility of direct visual contact of the industrial estates with this type of activity. Second, the relationship between the distance of the place of residence, work or study to the petrochemical industry is unclear: in the first

case, the highest perceptions occur among those who live nearby (less than 1 km away) and those who live furthest away (more than 10 km). Perhaps the explanation of this fact, in the first case, is because these people identify themselves as the first to be potentially affected in the event of an accident. In contrast, among the second group, it was precisely the high-risk perception that made moving away from the petrochemical industry an important factor when choosing a place of residence. On the other hand, but along the same lines, those who chose their place of residence because of the presence of the petrochemical industry perceive a low risk. Finally, the people who work closest to the two industrial estates with petrochemical activity have much lower risk perception values than the rest, clearly because of the over-representation within this group of people who work in the industrial sector.

**Table 4.** Distribution of the population surveyed according to their perception level of the petrochemical risk and its territorial variables.

| Variable | Category | Perceived Level of Risk | | | | | |
|---|---|---|---|---|---|---|---|
| | | Very Low (1) | Low (2) | Medium (3) | High (4) | Very High (5) | Average |
| Visibility | Yes | 3.5% | 10.0% | 24.2% | 40.7% | 21.6% | 3.64 |
| | No | 4.5% | 8.3% | 23.1% | 36.5% | 27.6% | 3.75 |
| Distance to the place of residence | Less than 1 km | 0.0% | 0.0% | 16.7% | 50.0% | 33.3% | 4.17 |
| | Between 1 and 2 km | 7.7% | 11.5% | 19.2% | 42.3% | 19.2% | 3.54 |
| | Between 2 and 5 km | 4.8% | 10.7% | 25.0% | 33.3% | 26.2% | 3.65 |
| | Between 5 and 10 km | 3.7% | 9.4% | 24.2% | 37.7% | 25.0% | 3.71 |
| | More than 10 km | 0.0% | 3.5% | 24.1% | 58.6% | 13.8% | 3.83 |
| Distance to the place of work/study | Less than 1 km | 22.2% | 27.8% | 22.2% | 16.7% | 11.1% | 2.67 |
| | Between 1 and 2 km | 0.0% | 4.2% | 29.2% | 50.0% | 16.7% | 3.79 |
| | Between 2 and 5 km | 2.9% | 8.6% | 22.9% | 34.3% | 31.4% | 3.83 |
| | Between 5 and 10 km | 4.0% | 9.7% | 24.2% | 37.2% | 24.9% | 3.69 |
| | More than 10 km | 0.0% | 0.0% | 12.5% | 50.0% | 37.5% | 4.25 |
| Importance of the petrochemical facilities in the choice of residence | Born here or arrived before | 1.3% | 6.7% | 24.1% | 38.8% | 29.0% | 3.88 |
| | No, I did not know it was there | 4.9% | 13.1% | 21.3% | 36.1% | 24.6% | 3.62 |
| | Yes, I came despite of the petrochemicals | 2.9% | 12.4% | 21.9% | 40.0% | 22.9% | 3.68 |
| | Yes, I came because the petrochemicals were here | 36.4% | 36.4% | 27.3% | 0.0% | 0.0% | 1.91 |
| Total | | 3.9% | 9.3% | 23.9% | 38.8% | 24.2% | 3.68 |

As can be observed (Table 5), the influence of the different independent variables over the fact of having a high perception of the petrochemical risk is very uneven:

- Age. If the age variable is entered into the model individually, its contribution is significant in statistical terms, in the sense that, the older the respondent, the greater the probability of having high-risk perception. However, in the multivariable model, its contribution becomes insignificant.
- Gender. As with age, gender stands out as an important variable: the probability that women have a high perception of the petrochemical risk is virtually double the figure among men.
- Level of education. Taking the population with secondary studies as a reference, it is clearly observed that the two remaining groups (people with professional training and university studies) have higher perception levels, particularly among the first group indicated.
- Employment status vis-à-vis the petrochemical industry. The fact of working directly in the industrial sector analyzed reduces the probability of having a high perception of the petrochemical risk by more than half.
- Visibility. This is the only territorial variable that reveals a significant statistical relationship with the high perception of risk: among the people who have visual

contact with the petrochemical facilities, the probability of a negative perception increases by 50 % with respect to those who say they have no direct visibility.

- Age of the residence and distances to the place of residence and work/study. The three territorial variables indicated appear insignificant, if analyzed separately and entered together with the rest of the variables in the model.

**Table 5.** Results of the logistics regression model.

| Variable | B | E.T | Wald | gl | Sig. | Exp(B) |
|---|---|---|---|---|---|---|
| Level of education | | | 3.784 | 2 | 0.151 | |
| Level of education (1) | 1.195 | 0.629 | 3.606 | 1 | 0.058 | 3.303 |
| Level of education (3) | 0.699 | 0.456 | 2.351 | 1 | 0.125 | 2.011 |
| Gender (1) | 0.578 | 0.254 | 5.190 | 1 | 0.023 | 1.783 |
| Employment (1) | −1.012 | 0.276 | 13.404 | 1 | 0.000 | 0.363 |
| Age | 0.013 | 0.013 | 1.024 | 1 | 0.312 | 1.013 |
| Distance to the place of residence | 0.000 | 0.000 | 0.309 | 1 | 0.578 | 1.000 |
| Distance to the place of work/study | 0.000 | 0.000 | 0.722 | 1 | 0.395 | 1.000 |
| Visibility | 0.359 | 0.269 | 1.790 | 1 | 0.181 | 1.432 |
| Age of the residence near | | | 5.822 | 4 | 0.213 | |
| Age of the residence near (1) | 0.196 | 0.632 | 0.096 | 1 | 0.756 | 1.217 |
| Age of the residence near (2) | 0.577 | 0.643 | 0.804 | 1 | 0.370 | 1.780 |
| Age of the residence near (3) | 0.658 | 0.601 | 1.200 | 1 | 0.273 | 1.931 |
| Age of the residence near (4) | 1.144 | 0.638 | 3.212 | 1 | 0.073 | 3.140 |
| Constant | −1.630 | 0.896 | 3.312 | 1 | 0.069 | 0.196 |

### 4.3. The Perceived Risk Level According to the Environmental Conditions

The levels of risk perception are noticeably lower when the general conditions are more favorable (during the hours of daylight and on sunny days), so, generally speaking, the visual reference with the petrochemical estate seems to be a source of tranquility (Table 6).

**Table 6.** Distribution of the surveyed population according to their perception level of the petrochemical risk and the environmental conditions.

| Environmental Conditions | Perceived Level of Risk | | | | | Average |
|---|---|---|---|---|---|---|
| | Very Low (1) | Low (2) | Medium (3) | High (4) | Very High (5) | |
| Day-time | 15.1% | 20.6% | 31.8% | 23.3% | 9.3% | 2.91 |
| Night-time | 8.8% | 13.2% | 23.9% | 32.9% | 21.1% | 3.44 |
| Sunny day | 14.2% | 20.4% | 33.2% | 23.2% | 9.0% | 2.93 |
| Rainy day | 10.4% | 16.0% | 25.8% | 28.8% | 19.0% | 3.30 |
| Flames and intense smoke | 4.9% | 5.8% | 13.9% | 28.5% | 46.9% | 4.07 |
| Intense noise | 4.9% | 8.1% | 14.6% | 30.9% | 41.5% | 3.96 |
| Intense bad smells | 1.6% | 5.1% | 8.8% | 26.2% | 58.2% | 4.34 |
| General risk | 3.9% | 10.2% | 24.1% | 36.9% | 24.8% | 3.68 |

On the other hand, consequently, uncertainty and concern are greater during the moments and episodes characterized, firstly, by the lack of visibility (night-time or rainy days) and, secondly, by the emission of flames or smoke, noise or bad smells. Out of the responses obtained, it can be deduced that this last factor is the one that concerns the surveyed population the most. This is a fact which, obviously, has to be linked to the fear

that this emission of bad smells is linked to the hypothetical episodes of emission gases that can lead to the formation of toxic clouds.

## 5. Discussion and Conclusions

PPGIS data quality is inextricably linked to sampling design and participation rates. The PPGIS frequently involves layperson or non-expert segments of society; thus, there may be questions about the spatial accuracy of the data. It is assumed that the PPGIS should be evaluated using the same spatial accuracy standards as expert GISs that map physical landscapes [23]. Nonetheless, PPGIS studies that have been conducted to assess the spatial accuracy of certain mapped attributes suggest that even in the mapping of physical landscape features, a non-expert public can achieve reasonable accuracy. For example, New Zealand residents were able to accurately map the location of native vegetation [24], while a general public sample in the United States was able to reasonably identify the locations of different wildlife habitats [25].

Using an integrated PPGIS in a survey on the population living or working near the petrochemical industry has proven to be an effective tool for studying the risk perception derived from this economic activity. Thus, as observed in the mapping results, the population clearly identifies the two petrochemical estates in Camp de Tarragona and its more immediate population hubs, as sources of risk. Furthermore, the risk due to explosion is considered to be the one that would have the greatest impact outside the industrial estates, with the western districts of the city of Tarragona, la Canonja, Vila-seca, La Pobla de Mafumet, El Morell, Vilallonga del Camp and Perafort being the most affected places.

It was expected that the perception of toxic cloud formation would have a wider scope; however, according to the population's perception, this risk is concentrated inside the industrial complex. As Cardona [26] explains, in the face of certain natural or anthropic phenomena, people have a fairly fragmented notion of risk perception, and this explains why from the technical perspective of some researchers, it is inadequate to define a society's acceptable risk level only by the perception of individuals, because unlikely yet sensational events tend to be perceived as more dangerous than more frequent and less recognized events. However, generally, people tend to underestimate risk more than overestimate it. Even so, 63 % of the population believe that their risk perception vis-à-vis the petrochemical industry is high or very high. Insofar as the relationship between the sociodemographic characteristics of the sample and the risk perception is concerned, the latter is higher in women than in men. Gender differences in risk perception have been described in various studies, with women reporting higher levels of risk as a source of concern more frequently than men. Women's greater sensitivity to and lower tolerance for risk is also common cultural wisdom [27,28]. As for age, the older people are, the greater their risk perception; therefore, it seems that as people grow older, they become more aware of possible dangers. This fact differs from the results obtained by Korstanje [29], who affirms that risk perception is greater in those who are economically active in comparison with other labor-passive groups, such as the under age, the retired or pensioners. Finally, the population that works or has direct family member employed in the petrochemical sector has a lower risk perception (50.3%) than the population who does not (67.4%). As Espluga [30] points out, this fact demonstrates that those who gain from this activity establish a much lower perceived risk level than the rest, and even cease to oppose this type of industry.

Regarding the distance between the industry and home, Glatron and Beck [31] affirm that the population living near hazardous industries is less aware of the industrial risk. This fact differs from the results obtained in this study, as the population residing nearer the petrochemical estates has a higher risk perception than the rest. In connection to this, it is possible that the accident that occurred in February 2020 in the firm, IQOXE, has increased the risk perception of the population living the closest to Polígono Sur. It is important to bear in mind that the expansive wave of the explosion broke windows, damaged façades and also affected the structure of some buildings, leaving the population without homes for several days and even killing one of the neighbors. The same scenario

occurred with the accident that took place in 2019 in Polígono Norte, where a worker died, although this accident did not have direct effects on the resident population beyond preventive confinement.

As for the environmental conditions, the population establishes a higher risk level at nighttime than during the day. This could be because the noises and light pollution the industry produces, increase with the calmness and darkness at night. The same occurs when it is a sunny day or rainy day. However, the risk perception increases more in exceptional situations or anomalies within the industry (flames, smoke, noises or bad smells), with bad smells being the episode that generates the highest sense of risk among the population, indeed, because of the possible effect that the gases generated in the petrochemical industry may have on people's health.

This study had some limitations as well. All PPGIS data contain potential bias from various sources, including participant geographic location, socio-demographic classes, participant beliefs, values, ideology and knowledge/experience in the study area [23]. This disparity can be explained by a "technologic gap" in which problems with accessing or comprehending geographic information technology make participation difficult for some users [15]. For example, in our study, nearly 59% of the sample was under the age of 34. However, this occurs as a result of the online method used. Additionally, the proportion of university graduates in the result of the survey was very high. This occurs because the university sent the link to all member of the academic world. However, we do not know whether the communication offices of factories, town halls, NGOs, etc., distributed the link to their partners or their communities. On the one hand, there is a generational bias linked to the digital bias and, on the other hand, it is not possible to control whether the survey reaches the entire population. Furthermore, the study had access restrictions for completing the survey since our PPGIS survey was only able to be conducted using a computer or laptop, not a smartphone or tablet.

The findings will be put to effective use by both the administration and chemical companies. These findings will also help to improve communication with the public in order to reduce the stress of potential risk. A burning torch, for example, is a safety measure, but people perceive it as a high-risk situation. Moreover, before progressing forward with future investigations, it is critical to determine whether the general public knows what to do in an emergency situation. In addition, participatory workshops and interviews will also be planned for future investigations to help understand how the population perceives risk. To determine whether there are any differences, the future study's case study scale could be increased, and a survey could be planned including the current study area and a control city, i.e., a city unaffected by the petrochemical industry.

To conclude, this study has succeeded in showing that the risk levels the population perceives are different according to gender, age and if they are direct or indirect beneficiaries of the petrochemical industry. From the territorial point of view, the population is fully aware of the location of the sources of risk and which areas will be directly affected in the event of an accident. Undoubtedly, this information will be useful for administrative purposes when communicating and managing the risk, and the PPGISs have proven to be an efficient tool for obtaining direct information on the risk perception of affected populations.

**Author Contributions:** All authors (E.B.P., J.A.G., Y.P.-A. and M.G.) contributed equally to conceptualizing, writing and reviewing this work. All authors have read and agreed to the published version of the manuscript.

**Funding:** This research was funded by (1) the Spanish Ministry of Science, Innovation and Universities (AEI/FEDER, UE) under Grant CHORA (contract number CSO2017-82411-P) and (2) Grant RESTAURA (contract number PID2020-114363GB-I00); (3) the GRATET Research Group, which is funded by the Catalan Government under code 2009-SG744; (4) the Agency for Management of University and Research (AGAUR, Generalitat de Catalunya, Spain) through 2017-SGR-245 grant; (5) Martí i Franquès program of Grants for Pre-Doctoral Research from Universitat Rovira i Virgili and Diputació de Tarragona under code 2020PMF-PIPF-16; and (6) the Spanish Ministry of Science and Innovation under Grant for predoctoral research training staff under code PRE-2021-098679. The

APC was funded by the Spanish Ministry of Science, Innovation and Universities (AEI/FEDER, UE) under Grant RESTAURA (contract number PID2020-114363GB-I00).

**Institutional Review Board Statement:** The study was conducted in accordance with the Declaration of Helsinki, and approved by the Ethics Committee of the UNIVERSITAT ROVIRA I VIRGILI (protocol code CEIPSA-2021-PR-0026 approved on 28 October 2022).

**Informed Consent Statement:** All subjects gave their informed consent for inclusion before they participated in the study. The study was conducted in accordance with the Declaration of Helsinki, and the protocol was approved on 28 October 2022 by the Ethical Committee Concerning Research into People, Society and the Environment of the Universitat Rovira i Virgili (CEIPSA-2021-PR-0026).

**Data Availability Statement:** Not applicable.

**Acknowledgments:** The authors wish to thank Marc Ajenjo Cosp, Researcher at the Centre d'Estudis Demogràfics, for his advice in constructing the logistic regression model.

**Conflicts of Interest:** The authors declare no conflict of interest. The funders had no role in the design of the study; in the collection, analyses, or interpretation of data; in the writing of the manuscript; or in the decision to publish the results.

## Appendix A

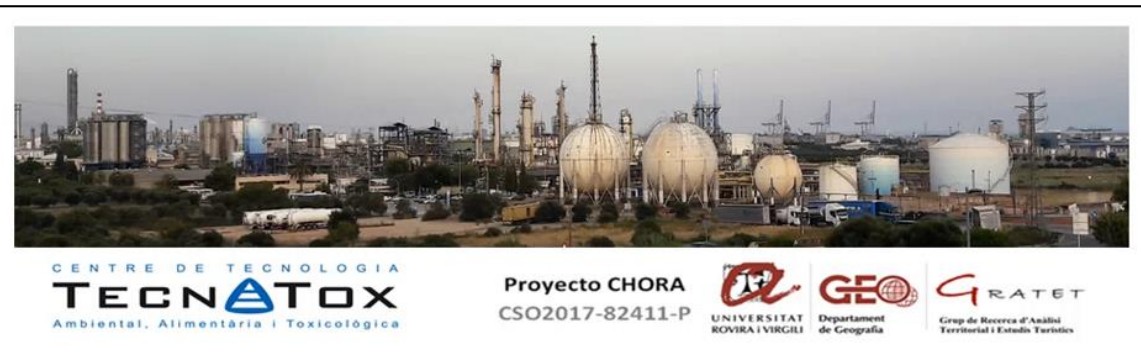

**Figure A1.** Information for respondents before starting the survey. Translation: Important: This survey must be conducted on a computer using Google Chrome or Mozilla Firefox. Not suitable for smartphones or tablets. Once answered, wait 10 min for another person to send a new response from the same device. This survey is included in the study of the impact of petrochemicals in the Camp de Tarragona that is carried out at the Universitat Rovira i Virgili. The data will be exclusively for academic use and the treatment there of will be in accordance with Organic Law 3/2018, of 5 December, protection of personal data and guarantee of digital rights.

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
