# Peer review of "Public Risk Perception of the Petrochemical Industry, Measured Using a Public Participation Geographic Information System: A Case Study of Camp de Tarragona (Spain)"

_environments, doi:10.3390/environments10030036_

Round 1

Reviewer 1 Report

This was a well written paper and I enjoyed reviewing it. I believe that the application of PPGIS was a suitable method to employ to address the public risk perception of the petrochemical industry in Camp de Taragona (Spain).

I have identified a number of issues which may need to be considered before publication of the article. Some of these are minor typos whereas others require some additional detail regarding the methods employed. That said, I think this paper is worthy of publication once these issues have been addressed.

1) There were a few issues with references in the text (see e.g. lines 39 and 43).

2) Line 46 - ...these types of risks...

3) Line 64 - proven not proved

4) Line 88 - as indicated by Grant?

5) Line 120 - proven not proved

6) Figure 1 - the place names are important and therefore I would like to see these clearly labelled on the map itself. Perhaps also consider including place names within Figure 6 as it is difficult to follow the descriptive text if you are not familiar with the area (e.g. lines 361-362).

7) Figure 3 - Typo - Example

8) Table 1 - the sample size was relatively small compared to the total population of the area (0.086%) and therefore I was wondering how the socio-demographics compared to the region as a whole. For example, nearly 59% of the sample were less than 34 years old. Was this as a result of the online method employed or does this reflect the population of the area as a whole?

9) How were respondents selected? Could anyone reply or was the sample supposed to reflect the socio-demographics of the area? More details are required on this so that the reader can interpret the result accordingly.

10) The proportion of University graduates seems very high - again was this as a result of the online PPGIS or the way the sample was collected? 

11) There is no discussion on the use of online PPGIS and whether this biases the sample.  I would have expected to see some discussion on this topic.

12) Have any other studies attempted to use PPGIS (or similar) to identify public perception of risk? This was not clear to me after reading the paper.

13) Figure 6 - the text is not readable within the figures - this needs to be checked before publication as it may just be low res images I have access to.

14) Figure 7 - it may be better to include the words (very low, low, moderate, high, very high) in the legend rather than 1-5 as it will make it easier for the reader to interpret.

15) Tables 4, 5 and 6 - headings should be above the table.

16) I am not an expert at statistics and therefore cannot comment on these elements within the paper - although what you have written makes sense to me as a non-expert.

17) Line 447 - becomes not be-comes

18) What level of information were the respondents supplied with regarding the petrochemical industry before they completed the questionnaire? This is important to state as it could influence their perception of risk. It may be worth reproducing the introductory text that was provided as an Annex?

19) Is it common within perceived risk studies that Females have a greater perception of risk that Males?

20) Table 6 - Day time vs Night-time - standardise whether they are hyphenated or not throughout.

21) What are the authors recommendations for future research in this field? It seems like a good approach to gain public perception data, but what do you think should be the next steps in developing the method or its application further? Points you may wish to consider include: What specifically could the results be used for and by whom? Are there issues with scaling the approach up? Should the results obtained be tested against other methods to test robustness in the findings? Could follow-up interviews be used to gain additional information to help explain the results?

Reviewer 2 Report

There needs to be extensive English usage editing -especially in the introduction.

line 197- what is "consistory"

Methodology

line 264- please explain why the 5 point Lickert scale was used for respondent risk assessment plus the statistical limitations involved

Results

lines 386-387 how was the 5 point Lickert scale converted to low vs. moderate vs. high risk perception? What was the rationale?

Discussion and Conclusions

Authors need to compare results to any other similar study using a similar method for a similar application. 

Also need to state study limitations, transferability and future research needed.

Reviewer 3 Report

The paper presents an interesting approach analyzing the population’s risk perception regarding the petrochemical industry, by applying a Public Participation Geographic Information System (PPGIS). Nevertheless, I have the following minor comments:

The ms should be proofread by a native speaker – minor spell check required (i.e. exemple etc.)

Figure 3. Exemple of a map-based question included in the survey – has a very bad quality and seems irrelevant as A and B can not be seen in figure 3 – also it is in Spanish.

Figure 4 – again the map itself does not show anything - it would be better to exclude it in order to better see the impact of possible risks in the area. – again it is in Spanish

Figure 7. Perception level of the petrochemical risk – please clearly indicate in the caption what does 1 – 5 mean – the reader should not go through all manuscript to be able to interpret the figure.

5. Discussion and Conclusions section should include further elaborations on the previous findings in relation to the existing ones and also what are the differences and your contributions (i.e.  Gavrilidis A.A. et al. (2022) Past local industrial disasters and involvement of NGOs stimulate public participation in transboundary Environmental Impact Assessment. Journal of Environmental Management 324: 116271, Patru-Stupariu I. et al., (2022) Impacts of the European Landscape Convention on interdisciplinary and transdisciplinary research. Landscape Ecology 37, 1211–1225, Nita A., et. al. (2022) Researchers’ perspective on the main strengths and weaknesses of Environmental Impact Assessment (EIA) procedures. Environmental Impact Assessment Review 92, 106690

In the reviewer's humble opinion, academics and industry practitioners will be even more interested in the effective ways to overcome any limitations in order to further improve on current practices and also increase yield (for example why use PPGIS and not other  method such as Social network analysis etc.).
